# Imbalanced Immune Response of T-Cell and B-Cell Subsets in Patients with Moderate and Severe COVID-19

**DOI:** 10.3390/v13101966

**Published:** 2021-09-30

**Authors:** Alexey Golovkin, Olga Kalinina, Vadim Bezrukikh, Arthur Aquino, Ekaterina Zaikova, Tatyana Karonova, Olesya Melnik, Elena Vasilieva, Igor Kudryavtsev

**Affiliations:** 1Almazov National Medical Research Centre, 197341 St. Petersburg, Russia; olgakalinina@mail.ru (O.K.); ikar19882012@gmail.com (V.B.); akino97@bk.ru (A.A.); catherine3452@yandex.ru (E.Z.); karonova@mail.ru (T.K.); orangelove@yandex.ru (O.M.); vasilieva_ey@almazovcentre.ru (E.V.); igorek1981@yandex.ru (I.K.); 2Institute of Experimental Medicine, 197376 St. Petersburg, Russia

**Keywords:** COVID-19, T-cell subsets, B-cell subsets, imbalanced immune response, multicolor flow cytometry

## Abstract

Background: The immunological changes associated with COVID-19 are largely unknown. Methods: Patients with COVID-19 showing moderate (*n* = 18; SpO2 > 93%, respiratory rate > 22 per minute, CRP > 10 mg/L) and severe (*n* = 23; SpO_2_ < 93%, respiratory rate >30 per minute, PaO_2_/FiO_2_ ≤ 300 mmHg, permanent oxygen therapy, qSOFA > 2) infection, and 37 healthy donors (HD) were enrolled. Circulating T- and B-cell subsets were analyzed by flow cytometry. Results: CD4+Th cells were skewed toward Th2-like phenotypes within CD45RA+CD62L− (CM) and CD45RA–CD62L− (EM) cells in patients with severe COVID-19, while CM CCR6+ Th17-like cells were decreased if compared with HD. Within CM Th17-like cells “classical” Th17-like cells were increased and Th17.1-like cells were decreased in severe COVID-19 cases. Circulating CM follicular Th-like (Tfh) cells were decreased in all COVID-19 patients, and Tfh17-like cells represented the most predominant subset in severe COVID-19 cases. Both groups of patients showed increased levels of IgD-CD38++ B cells, while the levels of IgD+CD38− and IgD–CD38− were decreased. The frequency of IgD+CD27+ and IgD–CD27+ B cells was significantly reduced in severe COVID-19 cases. Conclusions: We showed an imbalance within almost all circulating memory Th subsets during acute COVID-19 and showed that altered Tfh polarization led to a dysregulated humoral immune response.

## 1. Introduction

The novel coronavirus SARS-CoV-2 causes a disease named COVID-19 (derived from “coronavirus disease 2019”) by the World Health Organization (WHO). Its clinical manifestations may be associated with acute respiratory distress syndrome (ARDS) in severe cases of infection, leading to pulmonary oedema and respiratory failure as well as liver, heart, and kidney damage [1]. ARDS development is highly associated with the cytokine storm which manifests in typical clinical symptoms and increases the proinflammatory cytokine and chemokine levels in plasma. Activation and involvement of the adaptive immune system in the anti-SARS-CoV-2 immune reaction is one of the main processes leading to the development of specific antibodies, proinflammatory cytokines production, and the amplification of the antiviral response [2]. 

In most cases, lymphopenia with decreased absolute counts of CD4+ and CD8+ T cells, B cells, and natural killers (NK) cells have been detected in patients with COVID-19, excluding mild forms [3,4,5,6,7]. Decreased levels of total T cells as well as CD3+CD4+ and CD3+CD8+ subsets were primarily observed in elderly patients and were associated with infection severity and with the need for intensive care unit hospitalization [8]. Further, high expression of late activation markers (HLA-DR, CD38) was found on the T helper (Th) and cytotoxic T cells, along with an increased level of highly pro-inflammatory CCR6+ Th17 cells [4].

Thus, the levels of T cells and their main subsets, as well as the levels of B cells, could be crucial for COVID-19 development, prognosis, and outcomes [9]. However, investigations of the functional activity of T-cell subsets and the Th, cytotoxic T cells, and B cells’ repertoire in COVID-19 are limited. Clarification of the T-cell and B-cell subset ratios in patients with COVID-19 is necessary to understand the fine tuning of adaptive immune responses against SARS-CoV-2. Therefore, this study investigated the peripheral T-cell and B-cell subsets in patients with COVID-19, using multicolor flow cytometry.

## 2. Materials and Methods

### 2.1. Patient Characteristics

Forty-one patients with COVID-19 (16 men and 25 women) showing moderate or severe infection, who were admitted to the infectious disease department at Almazov National Medical Research Centre (St. Petersburg, Russia), and 37 apparently healthy donors (HD) (19 men and 18 women) were enrolled in this study. All the patients were admitted into the in-patient department in 5–7 days after illness onset. The median age of the patients was 61 (53;70) years, and they had a wide range of comorbidities (Table 1). The median age of HD was 37 (32:47) years. The research was approved by the local Ethics Committee of Almazov National Medical Research Centre (protocol No. 2209-20, 21 September 2020) and complied with the Helsinki Declaration. All patients included in the study provided informed consent.

The COVID-19 diagnosis was based on epidemiological and clinical data. All patients also underwent chest computed tomography (CT) and were tested for the presence of SARS-CoV-2 RNA in throat swab samples using a “SARS-CoV-2/SARS-CoV” PCR detection kit (DNA-technology TC, Moscow, Russia) and anti-SARS-CoV-2 IgM/IgG in plasma samples using a “DS-ELISA-anti-SARS-CoV-2” detection kit (RPC Diagnostic Systems, Nizhny Novgorod, Russia). Of the 41 patients, 32 had SARS-CoV-2 RNA in throat swabs as well as anti-SARS-CoV-2 IgM/IgG, whereas nine did not have detectable levels of SARS-CoV-2 RNA in throat swabs but were positive for the anti-SARS-CoV-2 IgM/IgG. 

The clinical and laboratory characteristics of the patients are presented in Table 2. According to the clinical data and severity of respiratory insufficiency, all the patients (*n* = 41) were divided into two groups: moderate (*n* = 18; SpO_2_ > 93%, respiratory rate > 22 per minute, body temperature > 38 °C, CRP level > 10 mg/L) and severe (*n* = 23; SpO_2_ < 93%, respiratory rate > 30 per minute, PaO_2_/FiO_2_ ≤ 300 mmHg, permanent need for oxygen therapy, qSOFA > 2) forms of COVID-19. 

### 2.2. Sample Collection

Blood samples were collected before treatment initiation. All investigations were performed less than 6 h after blood collection. Peripheral blood samples were collected into vacuum test tubes containing K3-EDTA anticoagulant and were then processed to analyze the relative and absolute numbers of the main T- and B-cell subsets by multicolor flow cytometry. A clinical blood analysis was performed using a Cell-DYN Ruby Hematology Analyzer (Abbott, Abbot Park, IL, USA). T-cell and B-cell immunophenotyping was performed by multicolor flow cytometry using a CytoFlex S Flow Cytometer (Beckman Coulter, Indianapolis, IN, USA). The flow cytometry data were analyzed using Kaluza software v2.1 (Beckman Coulter, Indianapolis, IN, USA).

### 2.3. T-Cell Immunophenotyping by Flow Cytometry

The Th-cell subsets were analyzed using multicolor flow cytometry. A whole peripheral blood (100 μL) sample was stained using FITC-labelled mouse anti-human CD45RA (clone ALB11, cat. IM0584U, Beckman Coulter, Indianapolis, IN, USA), PE-labelled mouse anti-human CD62L (clone DREG56, cat. IM2214U, Beckman Coulter, Indianapolis, IN, USA), PerCP/Cy5.5-labelled mouse anti-human CXCR5 (CD185, clone J252D4, cat. 356910, BioLegend, Inc., San Diego, CA, USA), PE/Cy7-labelled mouse anti-human CCR6 (CD196, clone G034E3, cat 353418, BioLegend, Inc., San Diego, CA, USA), APC-labelled mouse anti-human CXCR3 (CD183, clone G025H7, cat. 353708, BioLegend, Inc., San Diego, CA, USA), APC-Alexa Fluor 750-labelled mouse anti-human CD3 (clone UCHT1, cat. A94680, Beckman Coulter, Indianapolis, IN, USA), Pacific Blue-labelled mouse anti-human CD4 (clone 13B8.2, cat. B49197, Beckman Coulter, Indianapolis, IN, USA), and Brilliant Violet 510-labelled mouse anti-human CCR4 (CD194, clone L291H4, cat. 359416, BioLegend, Inc., San Diego, CA, USA). The staining protocols were performed according to the manufacturer’s recommendations. Briefly, 100 μL of whole peripheral blood was stained with the above-mentioned antibodies at room temperature for 15 min in the dark. After antibody staining, erythrocytes were lysed by adding 1 mL of VersaLyse Lysing Solution (Beckman Coulter, Inc., Indianapolis, IN, USA) with 25 μL of IOTest 3 Fixative Solution (Beckman Coulter, Inc., Indianapolis, IN, USA) in the dark at room temperature for 15 min. Next, all samples were washed (330x *g* for 8 min) twice with sterile PBS supplemented with 2% of fetal calf serum (FCS) (Sigma-Aldrich Co., Saint Louis, MO, USA), resuspended in 500 μL of fresh PBS with 2% neutral formalin (cat. HT5011-1CS, Sigma-Aldrich Co., Saint Louis, MO, USA), and subjected to a flow cytometry analysis. At least 40000 CD3+CD4+ Th cells were collected from each sample. 

Optimal combinations of antibodies conjugated with various fluorochromes were used according to a previously published method [10]. Staining was performed as described earlier. The gating strategy for the main T-cell subsets was described previously [11] and is shown in Figure 1. This method was developed by Sallusto et al. [12] and was validated by our group [13,14]. The hierarchical tree histogram in Figure 2 was gated on the CM Th and EM Th subsets as an example. 

Currently, there is no commonly accepted list of markers for the classification of Th subsets; however, we based our work on generally accepted recommendations from “Standardizing immunophenotyping for the Human Immunology Project”, published in 2012 [15] and 2016 [16], “Optimized Multicolor Immunofluorescence Panel 018: Chemokine receptor expression on human T helper cells” [17], as well as “Guidelines for the use of flow cytometry and cell sorting in immunological studies (second edition)”, published in 2019 [18]. However, many papers suggest that Th-cell subsets are not separate lineages but a continuum of mixed functional capacities, and cytokines from an inflammatory site microenvironment have a strong influence on Th-cell subsets’ “polarization” and result in so-called Th-cell “plasticity” that leads to alterations in transcription factor expression and cytokine production [19,20,21]. Conversely, the expression of chemokine receptors is strongly associated with skewing toward specific effector functions and the migratory behavior of different Th-cell subsets: CXCR3 facilitates the migration of Th1 cells to inflamed tissue sites along to gradients of chemokines CXCL9, CXCL10, and CXCL11 [22]; CCR4 on Th2 cells with CCL17 and CCL22 is critical for skin homing [23]; Th17 cells express CCR6 for migration to mucosal tissues that are enriched for CCL20 [24]; and, finally, CXCR5 allows Tfh cells to migrate from the T-cell zone into the B-cell follicles of lymph nodes that are enriched for CXCL13 [25]. However, taking into account that the chemokine receptors’ expression without measured cytokines levels or/and transcriptional factors expression indicated only the potential class of “polarized” T-helper cells, we named these subsets Th1-like cells, Th2-like cells, Th17-like cells, etc. 

### 2.4. B-Cell Immunophenotyping by Flow Cytometry

The frequency and phenotype of B cells in the peripheral blood of patients with COVID-19 and in the HD were analyzed by flow cytometry after staining 100 μL of whole peripheral blood with Alexa Fluor 488-labelled mouse anti-human IgD (clone IA6-2, cat. 348216, BioLegend, Inc., San Diego, CA, USA), PE-labelled mouse anti-human CD38 (clone LS198-4-3, cat. A07779, Beckman Coulter, Indianapolis, IN, USA), PC7-labelled mouse anti-human CD27 (clone 1A4CD27, cat. A54823, Beckman Coulter, Indianapolis, IN, USA), APC-labelled mouse anti-human CD24 (clone ALB9, cat. A87785, Beckman Coulter, Indianapolis, IN, USA), APC/Cy7-labelled CD19 (clone HIB19, cat. 302218, BioLegend, Inc., San Diego, CA, USA), Pacific Blue-labelled mouse anti-human CD5 (clone BL1a, cat. A82790, Beckman Coulter, Indianapolis, IN, USA), and Krome Orange-labelled mouse anti-human CD45 (clone J33, cat. A96416, Beckman Coulter, Indianapolis, IN, USA). An erythrocyte lysing procedure following by cell washing was performed as above-described. At least 5000 CD19+ B cells were analyzed for each sample. The gating strategy and algorithms used for the B-cell subset analysis are shown in Figure 3.

### 2.5. Statistical Analysis

A statistical analysis was performed using Statistica 7.0 (StatSoft, Tulsa, OK, USA) and GraphPad Prism 8 (GraphPad software Inc., San Diego, CA, USA) software. Normality was checked using Pearson’s chi-squared test. All data are presented as a percentage of positive cells in the main T- or B-cell subset subpopulation. The absolute numbers of immune cell subsets were calculated using the results of leukocytes and lymphocytes from the haematology analyzer. All the results are presented as median and interquartile range: Me (25;75). A dispersion analysis was performed using ANOVA statistics. The differences between groups were analyzed using the nonparametric Mann–Whitney U-test. Significance was set at *p* < 0.01.

## 3. Results

### 3.1. Routine Laboratory Tests

The median age differences between patients with moderate infection (62.5 (52;70) years) and patients with severe infection (59 (53;77) years) were not significant. Both groups had significantly lower levels of haematocrit, lymphocytes, eosinophils, and basophils compared to healthy donors (Table 2). Further, patients with severe infection showed decreased white blood cell and monocyte counts compared to the HD. All other routine laboratory tests demonstrated no differences between the patient groups. The CT results showed that patients with severe infection had significantly (*p* < 0.01) higher lung damage.

Routine laboratory tests were performed for patients with COVID-19 to diagnose the systemic inflammatory response syndrome (SIRS) and main organ dysfunction (Table 2). The levels of procalcitonin, D-dimer, and troponin I, were below the reference values at the beginning of the treatment. One of the main SIRS markers, CRP, was dramatically elevated to 64.9 (39.7;80.1) and 71.4 (36.1;109.5) in patients with moderate and severe infection, respectively. 

Haematology testing demonstrated no anemia or leukopenia in patients with COVID-19 (Table 2). Notably, lymphopenia (1.02 (0.74;1.26) × 10^9^/L) was observed in comparison with healthy donors, as well as compared to the region’s laboratory cut-off (1.09–2.99 × 10^9^/L).

### 3.2. Alterations in the Main T-Cell Subsets of Patients with COVID-19

To study the possible alterations of Th subsets, we first determined the relative number of CD3+CD4+ cells in the peripheral blood of both COVID-19 groups and in the healthy donors. The relative number of T-helper cells in patients with moderate (39.21 (33.69;47.77)%) and severe (38.31 (30.92;53.81)%) infection was significantly decreased (*p* < 0.001) compared to that in the healthy donors (55.66 (46.44;63.99)%). In both patient groups, absolute Th cell counts (488 (350;610) cells/mL and 495 (287;611) cells/mL, respectively), were also decreased (*p* < 0.001) compared with those in the HD (821 (676;922) cells/mL). Interestingly, neither the relative nor absolute Th counts were different in patients with moderate or severe COVID-19 infection.

Regarding the relative counts of circulating Th subsets at different maturation checkpoints, no significant differences were found in the central memory, immature (“naïve”), or TEMRA Th cells (Figure 4B–D). However, we found that the frequency of effector memory (EM) Th cells in patients with severe COVID-19 was significantly lower than that in patients with moderate COVID-19 and in the HD (10.28 (5.84;13.37)%, 13.00 (8.23;17.75)% and 15.65 (12.40;22.85)%), respectively (Figure 4A). Moreover, the relative number of EM Th cells demonstrated a significant decrease in patients with severe COVID-19 compared to that in patients with moderate infection. Simultaneously, both COVID-19 groups showed lower levels of Th cells in peripheral blood compared to the healthy donors (Figure 4E–G).

### 3.3. Imbalance of Main Polarized Th Cell Subsets in Patients with COVID-19

First, we analyzed the main Th cells within the central memory cells that reside in the secondary lymphoid organs and have the capacity to recirculate in peripheral blood. All patients and HD had similar levels of Th1-like cells in circulation, as shown in Figure 5. Patients with severe disease had higher levels of Th2-like cells compared to the moderate COVID-19 group and healthy donors (15.83 (13.40;22.40)% vs. 13.06 (12.30;14.76)%, *p* = 0.015 and 11.05 (9.93;13.44)%, *p* < 0.001, respectively) (Figure 5B). Further, we revealed a significantly lower proportion (*p* = 0.014) of CM Th17-like cells in patients with severe COVID-19 compared to HD (35.67 (28.47;43.26)% vs. 41.00 (37.31;46.13)%, respectively) (Figure 5C). Finally, we noted decreased values of CM Tfh-like cells in both COVID-19 groups (14.55 (11.67;18.01)%, *p* = 0.008 for moderate and 11.68 (10.14;14.28)%, *p* < 0.001 for severe infection) compared with HD (Figure 5D). Moreover, lower frequencies of Tfh-like cells were detected in patients with severe infection compared to those in patients with moderate infection (*p* = 0.021) (Figure 5D). 

Next, we investigated the alterations in Th subsets among effector memory cells that can migrate to inflamed non-lymphoid tissues and display different effector functions. As shown in Figure 5, both COVID-19 groups showed lower levels of Th17-like cells compared to the healthy donors (58.51 (46.94;65.73)% vs. moderate COVID-19–46.92 (25.12;61.60)%, *p* = 0.013 and severe COVID-19–49.55 (29.75;62.54)%, *p* = 0.036) (Figure 5G). However, there were no significant differences in the EM Th1-like, Th2-like, and Tfh-like cells between the groups of patients (Figure 5E–H).

Interestingly, among the patients with moderate infection the relative level of effector memory (EM) Th2-like cells was increased (*p* = 0.033), and the proportion of EM Th17-like cells was decreased (*p* = 0.005) in patients with arterial hypertension than in those without this comorbidity. It was the only significant difference demonstrating the possible influence of comorbidities on the T- and B-cell subsets during COVID-19.

### 3.4. Imbalance of Th17-Like Subsets in Patients with COVID-19

An analysis of these different CM Th17-like cell subsets (Figure 6) in severe COVID-19 patients indicated that the frequency of “classical” Th17-like cells was increased whereas the level of Th17.1-like cells was decreased compared to that in the HD (44.88 (43.45;39.91)% vs. 35.71 (27.72; 42.23)%, *p* = 0.001 and 18.54 (12.25;22.07)% vs. 27.36 (22.12;35.27)%, *p* < 0.001, respectively) (Figure 6A,B). We also found decreased levels of CM Th17.1-like cells in patients with moderate COVID-19 compared to HD (19.17 (15.56;23.08)% vs. 27.36 (22.12;35.27)%, *p* < 0.001) (Figure 6B). The relative numbers of DP Th17-like cells in the moderate COVID-19 group were higher than those in the healthy donors (32.18 (28.69;35.53)% vs. 28.47 (24.00; 33.15)%, *p* = 0.035) (Figure 6C). These were the only significant differences in CM Th-cell subsets of patients with moderate infection. 

The alteration in Th17-like subsets within EM Th17-like cells in patients with severe COVID-19 was very similar to the before-mentioned alteration within CM Th17-like cells. Patients showed increased levels of classical Th17-like cells, whereas the frequency of Th17.1-like cells was decreased compared to those in HD (Figure 6F). Regarding the levels of circulating EM Th17-like subsets, no significant differences were found between the moderate COVID-19 group and healthy donors. Further, CM Th17-like and EM Th17-like cell subsets did not demonstrate significant severity related differences between both patient groups.

### 3.5. Alterations in Tfh-Like Subsets in Patients with COVID-19

Considering the potential role of specific humoral immunity in the response to SARS-CoV-2 infection, we focused on circulating Tfh-like cells and their main subsets. It is well known that the expression of CXCR3 and CCR6 chemokine receptors allows the definition of four distinct Tfh-cell subsets corresponding to the polarized non-Tfh-cell subsets Th1, Th2, and Th17 cells [26]. Thus, a flow cytometric analysis of CXCR3 and CCR6 co-expression permitted the identification of Tfh1-like (CXCR3+CCR6−), Tfh2-like (CXCR3−CCR6−), Tfh17-like (CXCR3−CCR6+), and DP Tfh-like (CXCR3+CCR6+) cells. In this study, we screened the expression of the above-mentioned chemokine receptors on circulating central memory CXCR5+positive Th cells in patients with COVID-19 and healthy donors to identify circulating central memory Tfh-cell subsets (Figure 1G,J). First, we found no significant differences between the severe and moderate COVID-19 group as well as between the healthy donors and moderate COVID-19 group. Further, we revealed altered CM Tfh-like subsets in the severe COVID-19 group (Figure 7). We found that severe disease was associated with lower levels of Tfh1-like cells (22.66 (20.33;26.40)% vs. 28.74 (24.23;33.59)% in HD, *p* = 0.001) (Figure 7A). Furthermore, patients with severe COVID-19 showed a shift in circulating Tfh-like cells toward Tfh17 polarization compared to HD (40.42 (35.34;43.13)% vs. 33.76 (27.55;37.29)%, *p* = 0.001, respectively) (Figure 7C).

### 3.6. Alterations in Peripheral Blood B-Cell Subsets from Patients with COVID-19

As we found significant alterations in the Tfh-like subsets among patients with COVID-19, we analyzed circulating B cells because Tfh cells are known to control all stages of B-cell differentiation and activation that occur in peripheral lymphoid tissues [27]. We regarded the frequency of peripheral blood circulating B cells and found that both groups of patients with COVID-19 had significantly lower concentrations of CD19+ cells compared to those in the healthy donors (108.3 (76.0;195.1) cells/µL in severe COVID-19 and 84.9 (44.4;188.2) cells/µL in moderate COVID-19 vs. 208.9 (170.6;311.2) cells/µL in HD, with *p* > 0.001 and *p* > 0.001, respectively). Furthermore, the percentage of CD19+ cells in blood samples from patients with moderate COVID-19 was significantly lower than that in the healthy donors (8.7 (6.2;12.2)% vs. 11.4 (9.5;15.3)%, *p* = 0.020), whereas in patients with severe infection it was almost the same (11.1 (7.5;18.2)%).

Next, we estimated the frequency of different B-cell subsets in patients with COVID-19. The so-called “Bm1-Bm5” classification was used as one of the major classification schemes based on the relative expression of IgD and CD38 [28]. According to this scheme, IgD and CD38 staining reveals IgD+CD38– “naïve” Bm1 cells, IgD+CD38+ “activated naïve” Bm2 cells, IgD+CD38++ pre-germinal-center Bm2 cells, IgD-CD38++ so-called “Bm3 + Bm4” cells–the common subset, containing centroblasts and centrocytes, as well as two types of circulating memory B-cell subsets–IgD–CD38+ early memory and IgD–CD38− resting memory cells (eBm5 and Bm5 cells, respectively) (Figure 3E). The results are summarized in Table 3. We did not identify population differences between the COVID-19 groups. Meanwhile, both COVID-19 groups showed increased levels of Bm3 + Bm4 subsets, whereas the levels of Bm1 and Bm5 cells were lower than those in the healthy donors. The relative count of “activated naïve” Bm2 B cells in both COVID-19 groups was similar to that in HD, whereas the absolute numbers of these subsets were significantly lower.

To analyze the levels of memory B-cell subsets in peripheral blood, B cells were also classified using the IgD- vs. -CD27 expression system [29], which defines five main B-cell subsets including IgD+CD27– naïve B cells, IgD+CD27+ unswitched memory and IgD–CD27+ class-switched memory B cells, IgD–CD27− double-negative memory B cells, and IgD–CD27++-circulating plasma cell precursors (Figure 3I) (Table 4). The frequency of naïve B cells was similar in both COVID-19 groups and in the healthy donors, whereas the frequency of unswitched and class-switched memory B cells was significantly reduced in patients with severe COVID-19 compared to the HD. Moreover, the relative count of unswitched memory B cells was also higher in patients with severe disease than in those with moderate COVID-19. Finally, both groups of patients showed increased relative and absolute counts of plasma cell precursors in the peripheral blood compared to those in the HD.

## 4. Discussion

It was previously reported that patients with severe COVID-19 showed higher leukocyte and neutrophil counts, a higher neutrophil-to-lymphocyte ratio (NLR), and lower levels of monocytes, eosinophils, and basophils compared to those in patients with moderate disease severity [3,5,7,30,31,32]. In addition, a higher neutrophil count and NLR were associated with the severity of infection [30] and could be predictors of a poor prognosis [33]. In contrast, in our patients, we observed decreased levels of leukocytes to be most pronounced in severe infection with no significant differences in the neutrophil count compared to the healthy donors. Patients with moderate and severe infections in our study showed a decreased lymphocyte ratio. There were no significant differences in lymphocyte levels between the two groups of patients. The eosinophil and basophil counts were within the reference values and were lower than those in healthy donors.

Naïve T cells can leave the thymus after antigen-independent differentiation but do not undergo antigen-dependent differentiation in secondary lymphoid organs. CM T cells are less-differentiated, long-lived memory cells, with very limited effector functions. They recirculate through secondary lymphoid organs and demonstrate high proliferative and renewal capacity [13,14,34,35]. Central memory T cells produce high amounts of IL-2 and low levels of other effector cytokines (IL-4, IL-5, and IFN-γ). EM T cells represent more-differentiated circulating effector cells that can rapidly enter inflamed tissues due to the upregulated expression of tissue-homing chemokine receptors and adhesion molecules and provide a rapid and effective defense response [13,35]. The lack of co-stimulating molecules on TEMRA cells demonstrates that professional effector cell activation can be performed without additional co-stimulation [36]. TEMRA cells rapidly migrate to various anatomical sites to promote pathogen clearance by producing inflammatory cytokines and cytotoxicity. 

A recent study has reported a relative loss of peripheral blood CD45RA+CD27+CCR7+CD95− naïve CD4 T cells compared with the controls, but increased CD45RA−CD27−CCR7+ EM2 and CD45RA+CD27−CCR7− EMRA cells in patients with COVID-19 compared to the healthy donors [37]. Contrasting results were reported by De Biasi et al. who showed that patients with COVID-19 had similar relative counts of CD4+ T cells compared to the controls (with the exception of CCR7−CD45RA+CD28−CD27+/− terminal effector Th cells that were decreased compared to the control), but the absolute number of these cells was significantly lower [38]. Meanwhile, a study by Gutiérrez-Bautista J. et al. demonstrated that naïve CD4+ T cells (CD45RA+, CXCR3−, CCR4−, CCR6−, CCR10−) were greatly overrepresented in non-ICU hospitalized patients with COVID-19 and ICU hospitalized patients [39]. Our data indicated that in severe COVID-19, only the frequency of EM Th cells was decreased compared with that in the other groups, whereas the absolute numbers of all Th subsets were lower during SARS-CoV-2 infection. 

No significant differences in the differentiation of CD4 T cells were found between healthy donors and patients with mild COVID-19, but patients with severe disease showed an expansion of central memory CCR7+CD45RO+ CD4 T cells [40]. A high prevalence of central memory CD45RA−CD27+CD28+CCR7+ CD4 T cells in PBMCs was also detected by Saris et al. in patients with mild COVID-19 compared to healthy donors [41]. 

Previously, both helper T cells (CD3+CD4+) and cytotoxic T cells (CD3+CD8+) in patients with COVID-19 were found to be below normal levels, and this decline in helper T cells was more pronounced in severe cases [5]. In addition, CD4+ T cells in patients infected with SARS-CoV-2 showed higher levels of CD69, CD38, and CD44 compared to those in the healthy donors [42]. 

The frequency of IFNγ-producing Th cells was found to be decreased among the total Th cells in patients with severe COVID-19 compared to those in patients with moderate infection [43]. Further, the level of circulating Th cells capable of producing IL-6, GM-CSF, and IFNγ was reported to be significantly increased in patients with COVID-19, especially in patients with severe infection [44]. Generally, the increase in IFNγ producing Th cells can be explained as a Th1 response to viral infection, but we did not find significant changes in the CM and EM Th1-like cells responsible for the anti-intracellular pathogens, and these were preferably associated with a type 1 immune response both in patients with severe and moderate viral infection. Moreover, the relative numbers of IFNγ-producing Th17-like and Tfh-like cell subsets were decreased in patients with moderate and severe infections. The situation was aggravated by the presence of lymphopenia, indicating the suppression of humoral immune response regulation. 

Interestingly, in our study the relative number of Th2-like cells was increased in patients with severe infection. The increase in Th2-like cells was presented predominantly by CM Th cells. These findings were unexpected because the Th2 response basically reflects allergy or helminth invasion. In particular, helminthes are known to have potent immunomodulatory effects in the body through the activation of anti-inflammatory Th2 immune responses which are characterized by the production of IL-4, IL-5, IL-9, IL-10, and IL-13 [45]. Furthermore, the increased levels of Th2 cells and their cytokines were demonstrated to be crucial for the resolution of many infection diseases [46]. Patients who died from SARS showed a significant increase in the Th2 cytokines IL-4 and IL-5 [47]. Likewise, viral respiratory infections are the most common triggers of severe asthma exacerbation in children and adults [48]. Higher percentages of circulating Th2 cells were found in the patients with M.tuberculosis infection compared to the controls [49,50]. Within different autoimmune disorders, the majority of patients with asthma demonstrated a predominantly Th2 immune response [51]. Previous investigations have demonstrated that allergen provocation of the upper and lower respiratory tract inducing allergic airway inflammation leads to a significant decrease in ACE-2 expression, suggesting a possible relationship between allergic inflammation and a lower susceptibility to SARS-CoV-2 infection [48]. IL-13, the main Th2 cytokine, has been found to significantly reduce ACE-2 expression in both nasal and bronchial epithelium [48]. Considering that ACE-2 serves as the receptor for SARS-CoV-2, our findings of an increased Th2 response may suggest a possible protective mechanism against virus invasion. On the other hand, higher proportions of senescent CXCR3–CCR6− Th2 were associated with death in patients with COVID-19 [52]. Furthermore, the increased levels of Th2 cells as well as their hyperactivation were closely linked with gastrointestinal symptoms, including dyspnea, hyperperistalsis, and gastric fluid acidification in patients with COVID-19 [53]. Further, Th2 polarization in patients with COVID-19 may provoke type 3 hypersensitivity with the subsequent deposition of antigen-antibody complexes and the development of autoimmune illnesses [54]. It was also demonstrated that high levels of Th2 were associated with a bad prognosis and a fatal outcome of the COVID-19 disease [52].

Next, we found that within central and effector memory Th cells, pro-inflammatory Th17.1-like cells were decreased in patients with severe COVID-19, whereas the so-called classical Th17-like cells were increased (Figure 6). These data indicated that Th17-like cells presented a shift towards a pro-inflammatory phenotype that could be linked with disease severity. Our study highlights the importance of a Th17-like decrease associated with COVID-19 severity. In contrast with previous studies that identified high levels of Th17 cells in the peripheral blood of patients infected with SARS-CoV-2 [4,55,56], our data revealed that the frequency of Th17-like cells in patients with severe COVID-19 was significantly lower than that in the HD (Figure 5). In agreement with our data, De Biasi et al. showed that patients with COVID-19 displayed a lower level of Th17 cells with CCR6+CD4+CD3+ and CCR6+CD161+CD4+CD3+ phenotypes [38], and Gutiérrez-Bautista J. et al. demonstrated a persistently low frequency of markers associated with Th1, Th17, and Th1/Th17 memory-effector T cells compared to healthy donors [39]. However, De Biasi et al. also found an increased capability of CD4+ or CD8+ T cells for producing IL-17 after in vitro stimulation. Furthermore, it is well documented that IL-17A was significantly elevated in both moderate and severe cases of COVID-19 compared to the controls (reviewed in [57,58]).

We found that Th17.1-like cells were significantly increased within the total Th17-like subset in severe COVID-19 compared to those in moderate COVID-19 and in the HD. It has been shown that Th17.1 cells produce cytokines, including IL-17A, GM-CSF, IFN-γ, and TNF-α, which are associated with inflammatory conditions [59,60] and influence different cell types at the site of infection, including tissue macrophages, epithelial cells, and endothelial cells by inducing pro-inflammatory mediator expression and promoting neutrophil recruitment [61]. Thus, Th17.1 and their cytokines could be involved in secondary tissue damage during severe SARS-CoV-2 infection, but the role of Th17 cells in COVID-19 pathogenesis remains unknown. 

We found increased levels of Tfh-like cells in both COVID-19 groups. Despite the evaluated levels of circulating memory Tfh cells in both the lymph nodes and spleen of patients with COVID-19, Tfh cells with different phenotypes, CD4+ICOS+, CD4+ CXCR5+, and CD4+ Bcl-6+, were present at decreased levels [62]. Meanwhile, no difference was observed in the frequency of circulating CD45RA+CXCR5+ Tfh cells between healthy individuals and COVID-19 convalescent patients [63]. 

Our analysis of Tfh-like cell subsets in patients with COVID-19 showed an imbalance towards “pro-inflammatory” Tfh17-like cells that were closely linked with disease severity. Recently, the frequency of CXCR3+CCR6− cTfh1 and CXCR3–CCR6− cTfh 2 cells was found to be higher in COVID-19 convalescent patients than in healthy donors, whereas the levels of CXCR3–CCR6+ cTfh17 cells in COVID-19 convalescent patients were decreased [63]. 

Tfh cells can be characterized as a subset that facilitates T cell-dependent B-cell activity [64]. Only Tfh2 and Tfh17 subsets appeared to stimulate naïve B cells and could promote antibody production [26,64,65]. Furthermore, a decrease in Tfh1 subsets and an increase in Tfh2 and Tfh17 subsets among circulating memory Tfh cells could reflect an increase in efficient helpers that promote antibody generation in patients with autoimmune diseases [65]. 

Taken together our findings provide evidence that impaired Tfh-like cell differentiation in patients with COVID-19 could lead to the maturation of dysfunctional B cells and the alteration of the humoral immune response during acute COVID-19.

An imbalance in Tfh-cell subsets that are known to display distinct capacities to activate B cells could be linked to the abnormal distribution of the B cells. First, the absolute counts of almost all B-cell subsets were decreased in patients with COVID-19, demonstrating an imbalance of humoral immunity in general. Meanwhile, the increased CD19+ Bm3 + Bm4 subset was significantly higher in patients with moderate and severe infections, indicating, possibly, the active functioning of the germinal center. The increase in immature plasmablasts (IgD–CD27++ and CD27++CD38++) also corresponded to the disease status (Figure 8). Decreased levels of class-switched memory (IgD–CD27+) B cells were observed in patients with severe infection compared with those in patients with moderate infection indicating the relationship of these cells with COVID-19 severity.

In line with our observations, some authors have reported that decreased counts of activated B lymphocytes along with older age, high C-reactive protein, and impaired renal function are independent predictors of mortality in patients with COVID-19 [66,67,68]. Kaneko et al. [62] observed that germinal centers are lost in the lymph nodes and spleens in acute COVID-19, and that the proportions and absolute numbers of total CD19+ B cells in addition to naïve (IgD+CD27−), early transitional T1 and T2 (IgD+CD27−CD10+CD45RB−), and CXCR5+ follicular (IgD+CD27−CD10-CD73+) B-cell subsets were markedly reduced in severely ill patients with COVID-19 with high CRP levels as compared to convalescent patients and healthy donors. These data suggest a direct association between these B-cell phenotypic changes in peripheral blood and COVID-19 severity [62]. They also reported that increased proportions of activated naïve B cells, IgD-CD27−CXCR5− B cells, and plasmablasts in severely ill COVID-19 patients correlate with systemic inflammation and are specific for SARS-CoV-2. Within the IgD+CD27− compartment, patients with severe COVID-19 showed an increase in the number of disease-related presumed non-germinal-center-derived activated B cells compared to convalescent patients. These included activated naïve B cells (IgD+CD27−CD21loCD11-c^hi^) and atypical late transitional B cells (IgD+CD27-CD10-CD73-CXCR5−) [62]. All these statements are in concordance with our results and demonstrate the participation of B-cell subsets in the immune reactions to COVID-19 and disease severity.

**Limitations:** Most patients were older than 50 years of age. Due to the limited number of subjects older than 50 years lacking signs of any chronic disorder, the formation of the healthy donor group was problematic and led to significant differences in the median age of the compared groups.

## 5. Conclusions

Th2-like cells were predominant within central and effector memory Th cells in patients with severe COVID-19.The relative number of Th17-like cells was decreased within central and effector memory Th cells in patients with severe COVID-19.“Classical” Th17-like cells represented the most predominant Th17-like subset in patients with severe COVID-19, whereas the levels of “non-classical” Th17-like or Th17.1-like cells were dramatically decreased.The frequencies of circulating follicular Th-like cells were decreased within the central memory Th cells in patients with moderate and severe COVID-19, and Tfh17-like cells represented the most predominant Tfh-like subset in patients with severe COVID-19.Memory B cells were decreased and circulating plasma cell precursors were increased in the peripheral blood of patients with moderate and severe COVID-19.

## Figures and Tables

**Figure 1 viruses-13-01966-f001:**
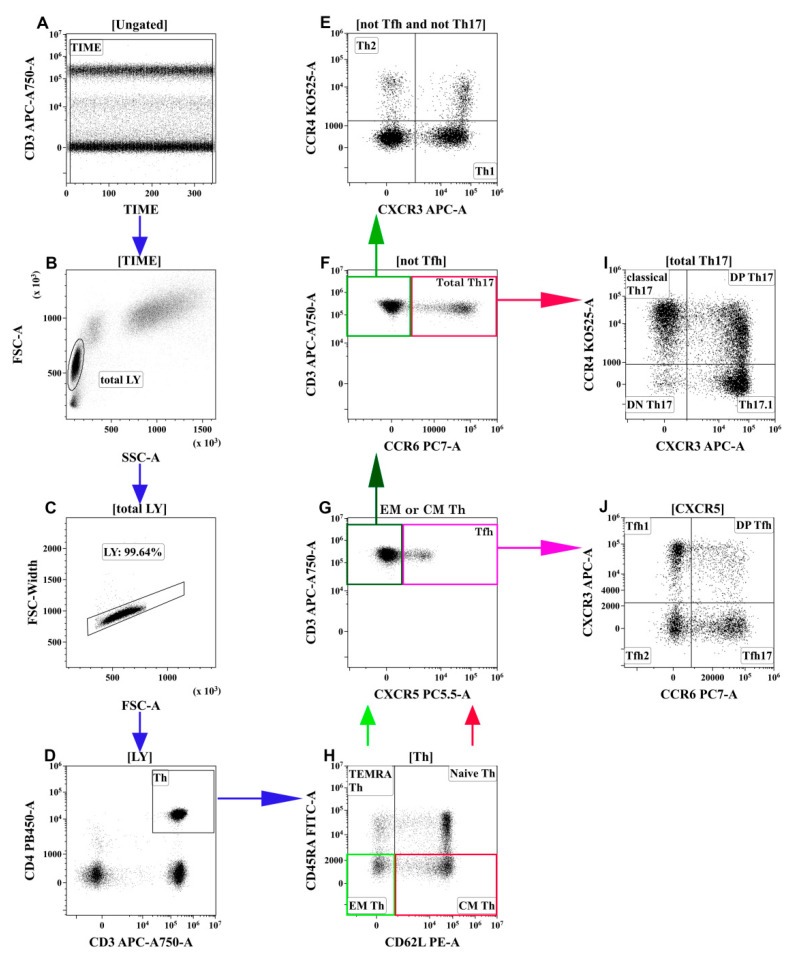
Flow cytometry immunophenotyping gating strategy for Th subsets. (**A**) Artifact exclusion included time gating. (**B**) FSC vs. SSC gating to discriminate lymphocytes and cell debris. (**C**) Doublet exclusions from the analysis using the FSC-area and FSC-width. (**D**) T-helper (Th) cells gating as CD3+CD4+ lymphocytes. Next, distinct Th-cell subsets were identified using different patterns of co-expressing antigens (**H**,**G**,**F**,**E**,**J**,**I**). (**H**) Th cells were identified as TEMRA (T effector memory re-expressing CD45RA, CD62L–CD45RA+), Naïve (CD62L+CD45RA+), EM (effector memory, CD62L-CD45RA−), and CM (central memory, CD62L+CD45RA–). Next, the latter two were analyzed independently using the following algorithm (**G**,**F**,**E**,**J**,**I**). (**G**) CM or EM Th cells were identified as Tfh-like (follicular Th) when they expressed CXCR5. (**J**) Tfh-like cells were classified as Tfh1-like (CCR6-CXCR3+), Tfh2-like (CCR6-CXCR3−), Tfh17-like (CCR6+CXCR3−), and DP Tfh-like (double-positive Tfh-like, CCR6+CXCR3+). (**F**) CXCR3-negative CM or EM Th cells were phenotyped by the expression of CCR6. CCR6-positive cells were identified as total Th17-like, which were classified as classical Th17-like (CXCR3-CCR4+), DN Th17-like (double-negative Th17-like, CXCR3-CCR4−), DP Th17-like (double-positive Th17-like, CXCR3+CCR4+), and Th17.1-like (CXCR3+CCR4−) (**I**). (**E**) The remaining CCR6-negative cells were identified as Th1-like (CXCR3+CCR4−) and Th2-like (CXCR3-CCR4+). Each colored arrow demonstrates the subsequent gating strategy for appropriate colored rectangle.

**Figure 2 viruses-13-01966-f002:**
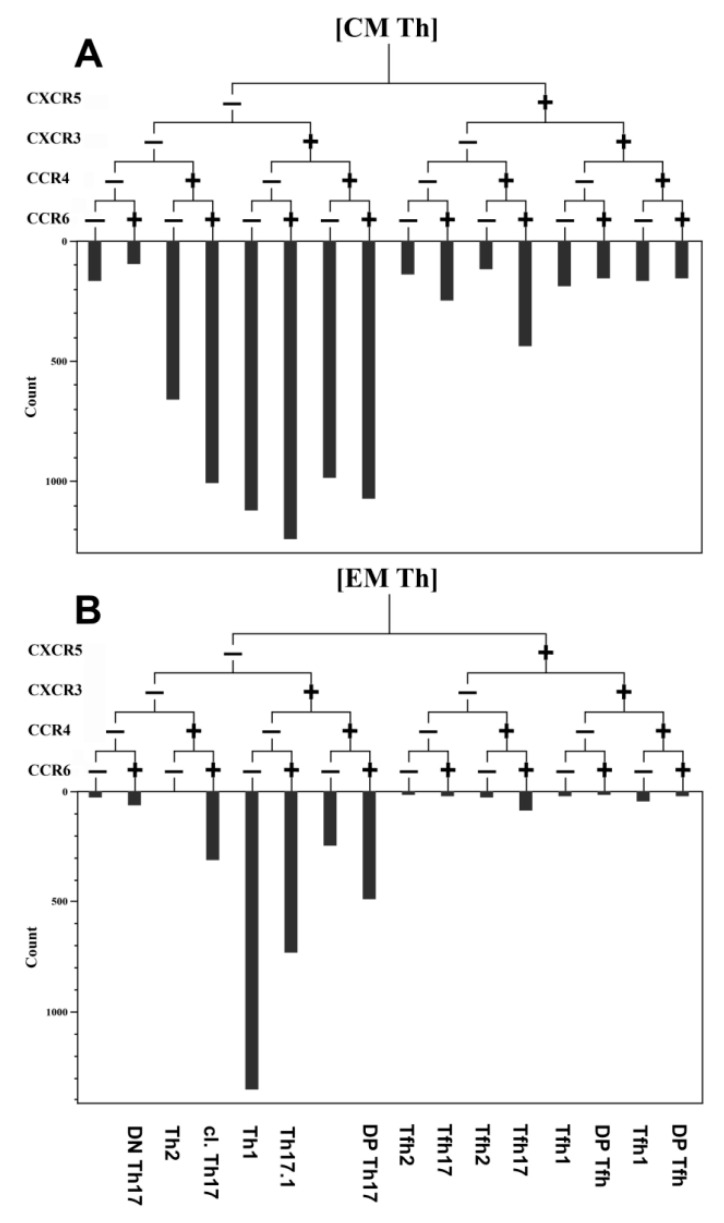
An example of the relative distribution of CM and EM Th cells subsets according to their expression of CXCR5, CXCR3, CCR6, and CCR4 in the healthy donor. (**A**) Central memory Th cells (CM). (**B**) Effector memory Th cells (EM). The frequency histogram below the trees indicates the relative proportion of cells with different patterns of CXCR5, CXCR3, CCR6, and CCR4 and is based on their expression in Th-cell subsets.

**Figure 3 viruses-13-01966-f003:**
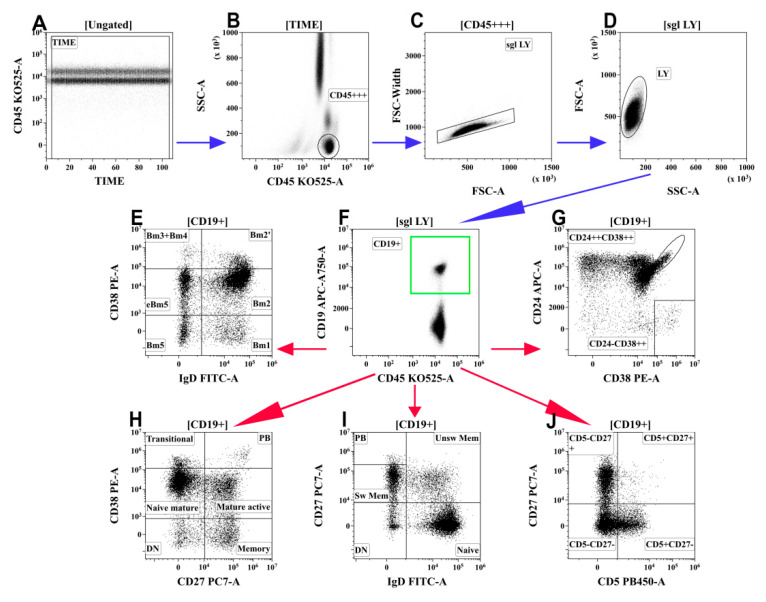
Gating and analysis strategy for B-cell immunophenotyping using flow cytometry. (**A**) Artifact exclusion included time gating. (**B**) Total lymphocyte subset purification based on side-scatter and bright CD45 expression. (**C**) Doublets exclusions from the analysis using FSC-area and FSC-width. (**D**) FSC; vs. SSC; gating to discriminate lymphocytes and cell debris. (**F**) B cells were gated as CD19+ lymphocytes. Next, distinct B cell subsets were identified using different patterns of co-expressing antigens (**E**,**G**–**J**). (**E**) Co-expression of IgD and CD38 (so-called “Bm1-Bm5” classification): six distinct B-cell subsets were identified, including “virgin naïve” Bm1 cells (IgD+CD38−), “activated naïve” Bm2 cells (IgD+CD38+), pre-germinal-centre Bm2′ cells (IgD+CD38++), common subset, containing centroblasts and centrocytes (so-called “Bm3 + Bm4” cells, IgD–CD38++), and early memory (eBm5) and resting memory cells (Bm5)–IgD–CD38+ and IgD–CD38−, respectively. (**G**) Co-expression of CD24 and CD38; transitional B cells were identified as CD24++CD38++ and circulating plasma cell precursors were identified as CD24-CD38++. (**H**) Co-expression of CD27 and CD38 allowed distinction of six B-cell subsets: CD27–CD38++ and CD27–CD38+ transitional and naive mature B cells, respectively; CD27–CD38− ‘double-negative’ B cell; CD27+CD38+ activated mature cells; CD27+CD38− resting memory cells; and circulating plasma cell precursors CD27++CD38++ (PB). (**I**) Co-expression of IgD and CD27 distinguished “naïve” B cells (IgD+CD27−) and three types of memory cells–“unswitched” memory cells (‘unsw mem’ IgD+CD27+), “class-switched” memory cells (sw mem IgD–CD27+), and so-called “double-negative” memory cells (DN IgD–CD27−), as well as circulating plasma cell precursors IgD–CD27++ (PB). (**J**) CD5 vs. CD27 co-expression allowed to identify the CD5+CD27− B-cell subset, enriched with regulatory B and B1 cells. Blue arrows mean common steps of each gating strategy for B-cells, red arrows mean different strategies for gating analysis of B-cells.

**Figure 4 viruses-13-01966-f004:**
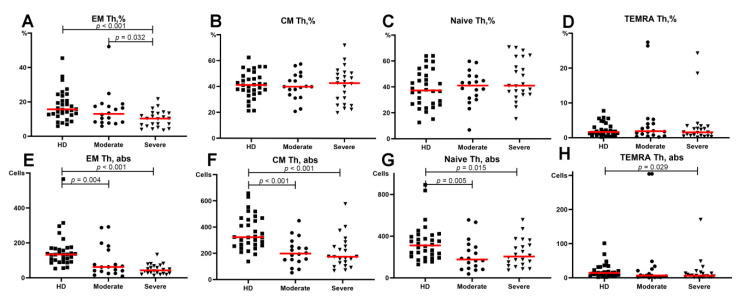
Basic Th cell subsets in patients with COVID-19 and in healthy donors. (**A**) Relative count of EM Th cells. (**B**) Relative count of CM Th cells. (**C**) Relative count of naïve Th cells. (**D**) Relative count of TEMRA Th cells. (**E**) Absolute number of EM Th cells. (**F**) Absolute number of CM Th cells. (**G**) Absolute number of naïve Th cells. (**H**) Absolute number of TEMRA Th cells. EM, effector memory; CM, central memory; TEMRA, T effector memory re-expressing CD45RA. Red lines demonstrate Median value.

**Figure 5 viruses-13-01966-f005:**
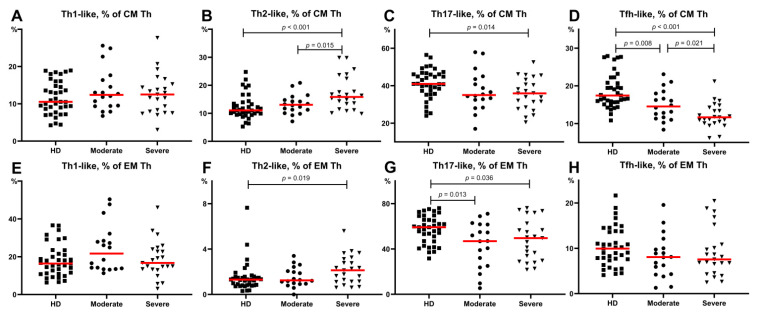
Th-cell subsets calculated from CM Th (**A**–**H**) or EM Th cells in patients with COVID-19 and in healthy donors. (**A**). Relative count of Th1-like cells calculated from CM Th cells. (**B**). Relative count of Th2-like cells calculated from CM Th cells. (**C**). Relative count of Th17-like cells calculated from CM Th cells. (**D**). Relative count of Tfh-like cells calculated from CM Th cells. (**E**). Relative count of Th1-like cells calculated from EM Th cells. (**F**). Relative count of Th2-like cells calculated from EM Th cells. (**G**). Relative count of Th17-like cells calculated from EM Th cells. (**H**). Relative count of Tfh-like cells calculated from EM Th cells. EM, effector memory; CM, central memory; Tfh, follicular Th-like cells. Red lines demonstrate Median value.

**Figure 6 viruses-13-01966-f006:**
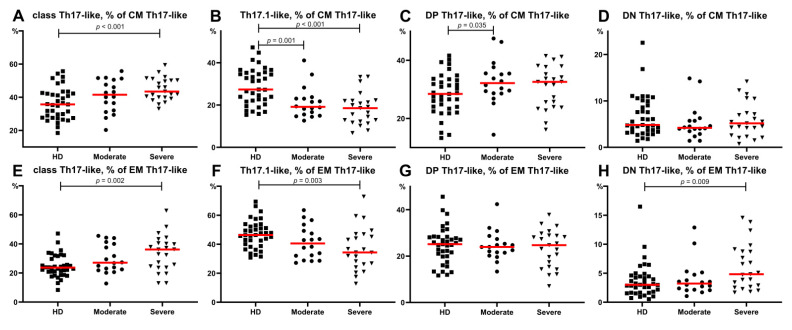
Th17-like cell subsets calculated from CM Th17-like (**A**–**H**) or EM Th17-like () cells in COVID-19 patients and in healthy donors. (**A**) Relative count of classical Th17-like cells calculated from CM Th17-like cells. (**B**) Relative count of Th17.1-like cells calculated from CM Th17-like cells. (**C**) Relative count of DP Th17-like cells calculated from CM Th17-like cells. (**D**) Relative count of DN Th17-like cells calculated from CM Th17-like cells. (**E**) Relative count of classical Th17-like cells calculated from EM Th17-like cells. (**F**) Relative count of Th17.1-like cells calculated from EM Th17-like cells. (**G**) Relative count of DP Th17-like cells calculated from EM Th17-like cells. (**H**) Relative count of DN Th17-like cells calculated from EM Th17-like cells. DP, double positive Th17-like (expressing CXCR3 and CCR4); DN, double negative Th17-like (lacking CXCR3 and CCR4). Red lines demonstrate Median value.

**Figure 7 viruses-13-01966-f007:**
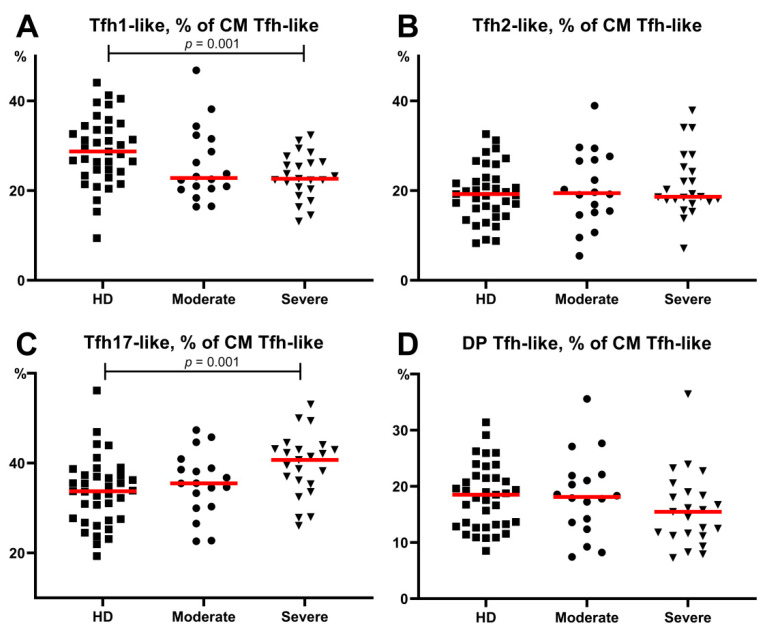
Relative count of Tfh-like cell subsets, calculated from CM Tfh-like cells. (**A**) Relative count of Tfh1-like cells. (**B**) Relative count of Tfh2-like cells. (**C**) Relative count of Tfh17-like cells. (**D**) Relative count of DP Tfh-like cells. CM, central memory; Tfh, follicular Th-like cells; DP, double positive Th17-like cells (expressing CXCR3 and CCR4). Red lines demonstrate Median value.

**Figure 8 viruses-13-01966-f008:**
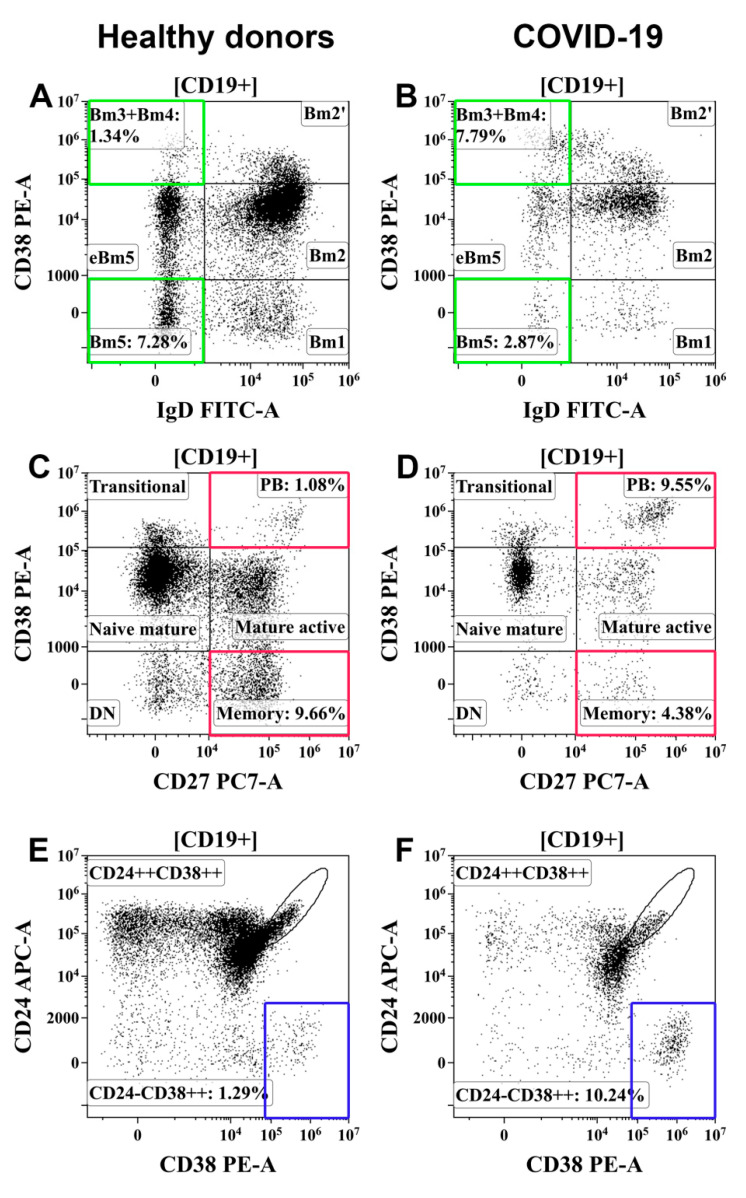
Examples of changes in B-cell subsets in patients with COVID-19. (**A**,**C**,**E**) Healthy donors. (**B**,**D**,**F**) Patients with COVID-19. (**A**,**B**) Bm3 + Bm4 and Bm5 B-cell subsets. (**C**,**D**) Resting memory B cells (CD27+CD38−) and circulating plasma cell precursors (CD27+CD38++) (PB). (**E**,**F**) Circulating plasma cell precursors (CD24−CD38++). Colored boxes highlight target B-cell subpopulations in patients with COVID-19 and healthy donors.

**Table 1 viruses-13-01966-t001:** Comorbidities of patients with COVID-19.

Comorbidities	Total Number of Patients, *n* (%)	Moderate, *n* (% Inside Group)	Severe, *n* (% Inside Group)
Arterial hypertension	19 (46)	7 (39)	12 (52)
Diabetes mellitus	5 (12)	2 (11)	3 (13)
Ischemic heart disease	9 (22)	4 (22)	5 (22)
Cancer in anamnesis	3 (7)	3 (17)	0
Bronchial asthma	5 (12)	2 (11)	3 (13)

**Table 2 viruses-13-01966-t002:** Characteristics of healthy donors and patients with COVID-19, Me (25;75). Cut-off values are given for an appropriate laboratory.

Parameter, Units (Reference Value)	Healthy Donors	Patients with COVID-19	
		Moderate (*n* = 18)	Severe (*n* = 23)
Age	37.0 (32.0;47.0)	62.5 (52.0;70.0) ^	59.0 (53.0;77.0) ^
CT, % lung damage		40 (32;52) *	60 (48;68) *
CRP, mg/L (0–5.0)		64.9 (39.7;80.1)	71.4 (36.1;109.5)
D-dimer, µg/mL (0–5.0)		0.55 (0.36;1.25)	0.62 (0.40;1.48)
Ferritin, ng/mL (30–400)		372.8 (148.4;901.1)	422.6 (164.5;713.4)
Fibrinogen, g/L (1.9–4.3)		6.0 (4.1;6.4)	6.0 (4.7;7.0)
Troponin I, ng/mL (0–0.034)		0.006 (0.002;0.014)	0.002 (0.002;0.002)
Procalcitonin, ng/mL (0–0.5)		0.07 (0.04;0.15)	0.06 (0.05;0.18)
ALT, U/L (0–41)		39.1 (25.2;72.1)	29.0 (23.0;63.5)
AST, U/L (0–40)		46.0 (31.0;80.0)	37.8 (32.0;61.0)
LDH, U/L (135–225)		335 (325;349)	332 (267;410)
Haemoglobin, g/L	139.5 (136.0;148.0)	128.6 (123.0;140.3)	141.2 (127.2;148.8)
RBC, ×10^12^	5.00 (4.70;5.13)	4.71 (4.07;5.02)	4.71 (4.45;4.91)
Ht, %	43.2 (41.5;46.0) *	38.6 (34.9;42.4) ^	40.2 (36.8;42.2) ^
Plt, ×10^9^	208 (194;235)	174 (153;212)	195 (130;301)
WBC, ×10^9^	5.8 (5.4;7.7)	5.1 (3.2;7.3)	4.4 (3.5;6.1) ^
Lymphocytes, ×10^9^	1.95 (1.78;2.21) *	1.03 (0.62;1.30) ^	1.02 (0.75;1.26) ^
Monocytes, ×10^9^	0.58 (0.39;0.71)	0.51 (0.29;0.58)	0.38 (0.24;0.51) ^
Neutrophils, ×10^9^	2.96 (2.78;4.31)	3.53 (1.98;4.65)	3.06 (2.31;3.97)
Eosinophils, ×10^9^	0.11 (0.08;0.23) *	0.02 (0.01;0.03) ^	0.01 (0;0.04) ^
Basophils, ×10^9^	0.08 (0.06;1.00) *	0.03 (0.02;0.04) ^	0.01 (0.01;0.04) ^

* *p* < 0.01, compared severe and moderate COVID-19. ^ *p* < 0.01, compared with HD.

**Table 3 viruses-13-01966-t003:** Absolute and relative numbers of B-cell subsets identified using the “Bm1-Bm5” classification in patients with COVID-19 and healthy donors.

Parameter	Healthy Donors		COVID-19 Patients		
	#	%	Severity	#	%
Bm1	22.2 (13.2;38.8)	10.7 (7.9;13.8)	Moderate	5.0 (3.4;7.7) *	6.1 (4.2;8.8) *
			Severe	4.2 (2.0;8.6) *	4.5 (1.6;6.4) *
Bm2	127.1 (91.8;184.2)	58.4 (52.6;63.7)	Moderate	40.0 (20.8;75.40) *	53.6 (44.2;61.8)
			Severe	73.7 (35.0;111.1) *	55.4 (48.3;68.9)
Bm2′	19.3 (11.2;25.1)	8.2 (6.4;10.3)	Moderate	8.2 (5.0;14.4) *	10.8 (8.0;14.4)
			Severe	13.8 (7.3;21.8)	11.2 (7.4;16.7) *
Bm3 + Bm4	2.6 (1.8;4.1)	1.1 (0.8;2.1)	Moderate	6.7 (3.8;13.3) *	10.5 (6.9;12.3) *
			Severe	6.1 (2.9;12.0) *	7.6 (3.0;15.2) *
eBm5	25.1 (16.4;37.4)	11.3 (8.7;14.1)	Moderate	9.3 (5.7;14.9) *	11.3 (7.9;15.7)
			Severe	8.4 (5.9;15.4) *	9.1 (5.6;12.1)
Bm5	15.9 (9.9;24.0)	7.2 (5.1;10.6)	Moderate	3.6 (2.3;4.9) *	5.5 (3.8;10.0) *
			Severe	3.8 (1.8;6.0) *	3.3 (1.8;7.1) *

# cells/µL of whole blood. % percentage within total CD19+ cells. * *p* < 0.01 compared with the appropriate parameters of HD, statistical analysis was performed using the Mann–Whitney U test.

**Table 4 viruses-13-01966-t004:** Absolute and relative numbers of B-cell subsets identified using IgD− vs. −CD27 classification in patients with COVID-19 and in healthy donors.

Parameter	Healthy Donors		COVID-19 Patients		
	#	%	Severity	#	%
Naïve, IgD+CD27–	139.5 (89.6;205.5)	65.4 (55.8;73.4)	Moderate	49.3 (20.5;87.8) *	56.1 (46.7;65.4)
			Severe	80.5 (35.1;127.7) *	62.9 (49.2;83.3)
Unswitched memory, IgD+CD27+	27.8 (17.9;42.2)	14.3 (9.8;17.9)	Moderate	9.2 (4.4;15.1) *	11.7 (7.1;19.9)
			Severe	7.8 (4.1;11.6) *	5.6 (4.0;14.0) *
Class-switched memory, IgD–CD27+	35.5 (25.6;52.3)	16.3 (11.5;20.4)	Moderate	13.2 (7.5;21.9) *	16.4 (11.9;23.3) ^
			Severe	9.5 (5.8;22.8) *	13.1 (5.6;16.6) *,^
Double-negative memory, IgD–CD27−	7.2 (5.6;12.6)	3.9 (2.3;4.8)	Moderate	4.1 (3.1;9.5) *	5.8 (4.4;8.1) *
			Severe	3.9 (2.5;11.3) *	4.3 (2.6;8.5)
Plasma cell precursors, IgD–CD27++	0.8 (0.3;3.3)	0.4 (0.1;1.8)	Moderate	3.3 (2.5;9.0) *	5.9 (3.5;8.1) *
			Severe	4.1 (2.0;10.4) *	4.9 (1.4;12.0) *

# cells/µL of whole blood. % percentage within total CD19+ cells. * *p* < 0.01 compared with the appropriate parameter of HD, statistical analysis was performed using the Mann–Whitney U test. ^ *p* = 0.02 compared with severe and moderate COVID-19, statistical analysis was performed using the Mann–Whitney U test.

## Data Availability

Not applicable.

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
