# Peer review of "Imbalanced Immune Response of T-Cell and B-Cell Subsets in Patients with Moderate and Severe COVID-19"

_viruses, 2021, doi:10.3390/v13101966_

Round 1

Reviewer 1 Report

Remarks for authors

  1. CRP >10mg/L does not differentiate patients with COVID-19 showing moderate from severe (it is the same in both groups).
  2. The median age of the patients was 61 (53;70) but the median age of HD was 37 (32:47) years. There is significant difference. Indeed the median age differences between patients with moderate infection (62.5 (52;70) years) and patients with severe infection (59 (53;77) 85 years) were not significant.

The most important similarity among these infectious diseases is the demographic affected most severely .This demographic tends to be the elderly, with comorbidities and degraded/dysfunctional immune systems, and others with degraded/dysfunctional immune systems. While there is some decline in the immune system with age, comorbidity is a stronger predictor of impaired immunity than chronological age in older adults

 It  is well known that expression of CXCR3 and CCR6 chemokine receptors allows the def346 inition of four distinct Tfh cell subsets corresponding to the polarized non-Tfh cell sub347 sets Th1, Th2, and Th17 cells (page 11 3.5 344 – 347) No References

Author Response

We thank reviewer for the attention to our work and for comments and questions. We performed some corrections in the manuscript according to reviewers recommendations. We hope that it will improve the quality of the manuscript. 

Remarks for authors

  1. CRP >10mg/L does not differentiate patients with COVID-19 showing moderate from severe (it is the same in both groups).

In this case CRP serve as a marker of severity of systemic inflammatory response syndrome. Whereas we formed patient’s groups according the severity of the COVID-19-associated pneumonia. Probably that is why CRP levels in groups of patients were not significantly different.

  1. The median age of the patients was 61 (53;70) but the median age of HD was 37 (32:47) years. There is significant difference. Indeed the median age differences between patients with moderate infection (62.5 (52;70) years) and patients with severe infection (59 (53;77) 85 years) were not significant.

Previously it was shown that number of T-cell subsets in healthy donors can be age related [1][2][3]. Thus, we agree with reviewer pointing out the significance of the age similarity in different groups. Understanding this we did our best to form the group of healthy donors as close as possible to groups of patients. There were two main problems: 1. the number of subjects older than 50 years lacking the signs of any chronic disorder is limited; 2. availability of persons older than 50 years was decreased because of lockdown limitations. Thus, the median age of patients and healthy donors was significantly different. However, realizing possible importance of this condition in the study design, we added the notification in the limitations section.  Besides, the significant differences in some studied parameters between moderate and severe patient groups indicate the value and validity of the obtained results.

The most important similarity among these infectious diseases is the demographic affected most severely .This demographic tends to be the elderly, with comorbidities and degraded/dysfunctional immune systems, and others with degraded/dysfunctional immune systems. While there is some decline in the immune system with age, comorbidity is a stronger predictor of impaired immunity than chronological age in older adults

We also agree with this statement. Thus, we included only persons without comorbidities or chronic disorders in the healthy donors group. We can declare that the significant differences between patients and donors are not related with comorbidities of the later.

 It  is well known that expression of CXCR3 and CCR6 chemokine receptors allows the def346 inition of four distinct Tfh cell subsets corresponding to the polarized non-Tfh cell sub347 sets Th1, Th2, and Th17 cells (page 11 3.5 344 – 347) No References

Thank you for your remark. We added appropriate reference.

Correction.

One reference was added.

26.          Morita R, Schmitt N, Bentebibel SE, Ranganathan R, Bourdery L, Zurawski G, et al. Human Blood CXCR5+CD4+ T Cells Are Counterparts of T Follicular Cells and Contain Specific Subsets that Differentially Support Antibody Secretion. Immunity [Internet]. 2011 Jan;34(1):108–21. Available from: https://linkinghub.elsevier.com/retrieve/pii/S1074761311000100

References

  1. Uciechowski P, Kahmann L, Plümäkers B, Malavolta M, Mocchegiani E, Dedoussis G, et al. TH1 and TH2 cell polarization increases with aging and is modulated by zinc supplementation. Exp Gerontol [Internet]. 2008 May;43(5):493–8. Available from: https://linkinghub.elsevier.com/retrieve/pii/S0531556507002732
  2. Xu D, Wu Y, Gao C, Qin Y, Zhao X, Liang Z, et al. Characteristics of and reference ranges for peripheral blood lymphocytes and CD4+ T cell subsets in healthy adults in Shanxi Province, North China. J Int Med Res [Internet]. 2020 Jul 10;48(7):030006052091314. Available from: http://journals.sagepub.com/doi/10.1177/0300060520913149
  3. Botafogo V, Pérez-Andres M, Jara-Acevedo M, Bárcena P, Grigore G, Hernández-Delgado A, et al. Age Distribution of Multiple Functionally Relevant Subsets of CD4+ T Cells in Human Blood Using a Standardized and Validated 14-Color EuroFlow Immune Monitoring Tube. Front Immunol. 2020;11(February):1–16.

Reviewer 2 Report

Although the manuscript has a good quality to be published, some points should be taken into consideration before that:

Methodology

  • The methodology was correctly used. Monoclonal antibody panels were correctly applied as a strategy to identify the different subpopulations studied, but it should be mentioned that other strategies had been applied in the literature that could affect the obtained results. Paragraph in lines 83-90 should be included in the results.
  • There were not included data about days after disease onset that could influence the significance of the obtained results and should be analyzed.
  • Signs “+” and “-“ in Figure 1 should be increased in size for a better visualization.
  • Table 2: First column header should be corrected to Parameter, units (Reference value), that reflects better the way that data was shown in Table 2.
  • Comorbidities should be independently shown for both COVID groups in Table 1, as they were analyzed and interpreted in results.

Results

  • Paragraph in lines 236-239 should be omitted because the strategy was explained in Material and Methods. Sentence in lines 139-140 should be omitted in Results and mentioned in methodology.
  • Paragraph in lines 261-268 and in lines 305-311 should also be omitted as it was explained in Material and Methods.

Discussion

  • Sentence in lines 483-484 is too categorical. It should only identified that other clinical situations have a Th2 response as those mentioned in the manuscript.
  • Sentence in lines 486-493 do not justify the severity observed in COVID-19 in relation to a polarized Th2 response so, it should be changed or omitted.
  • Sentence in lines 507-508: it should be better to express T cell populations as CCR6+ and CD4+CCR6+CD161+ T cells.
  • Omit sentence in lines 524-525 as other studies have shown differences in Tfh cells.
  • Omit sentence 543-545 because it is repetitive.
  • It should be recommended to mention future directions to be addressed at the end of the manuscript.

Author Response

We thank reviewer for the attention to our work and to valuable comments and questions. It gave us the opportunity to perform some corrections that can increase the quality of the manuscript. 

Methodology

  • The methodology was correctly used. Monoclonal antibody panels were correctly applied as a strategy to identify the different subpopulations studied, but it should be mentioned that other strategies had been applied in the literature that could affect the obtained results.

Corrections and clarifications were added to the manuscript.

Addition.

Currently, there is no commonly accepted list of markers for classification of Th subsets, but in our work we based on generally accepted recommendations from “Standardizing immunophenotyping for the Human Immunology Project”, published in 2012 [4] and 2016 [5], “Optimized Multicolor Immunofluorescence Panel 018: Chemokine receptor expression on human T helper cells”[6],  as well as “Guidelines for the use of flow cytometry and cell sorting in immunological studies (second edition)”, published in 2019 [7]. However, many papers suggest that Th cell subsets are not separate lineages but a continuum of mixed functional capacities, as well as cytokines from inflammatory site microenvironment have a strong influence on Th cell subsets “polarization” and result in so-called Th cell “plasticity” that leads to alterations in transcription factors expression and cytokine production [8][9][10]. Conversely, the expression of chemokine receptors is strongly associated with skewing toward specific effector functions and migratory behavior of different Th cell subsets: CXCR3 facilitates the migration of Th1 cells to inflamed tissue sites along to gradients of chemokines CXCL9, CXCL10, and CXCL11[11]; CCR4 on Th2 cells with CCL17 and CCL22 is critical for skin homing [12]; Th17 cells express CCR6 for migration to mucosal tissues that are enriched for CCL20[13]; and, finally, CXCR5 allows Tfh cells to migrate from the T cell zone into B cell follicles of lymph nodes that are enriched for CXCL13 [14].

  • Paragraph in lines 83-90 should be included in the results.

These sentences were translocated in the results section. 

  • There were not included data about days after disease onset that could influence the significance of the obtained results and should be analyzed.

The clarification was added to the “Patient Characteristics” section.

Addition.

All the patients were admitted in the in-patient department in 5-7 days after illness onset.

  • Signs “+” and “-“ in Figure 1 should be increased in size for a better visualization.

We are confused because there are no “+” and “-“ signs in the Figure 1. Probably reviewer was mistaken keeping in mind not Figure 1 but Figure 2. Figure 2 was corrected.

  • Table 2: First column header should be corrected to Parameter, units (Reference value), that reflects better the way that data was shown in Table 2.

Appropriate corrections were performed in the table 2.

  • Comorbidities should be independently shown for both COVID groups in Table 1, as they were analyzed and interpreted in results.

Corrections and clarifications were added in the Table 1.

Results

  • Paragraph in lines 236-239 should be omitted because the strategy was explained in Material and Methods.

These lines were deleted because of the gating strategy was described in details in the materials and methods section and in the Figure 1 legend.

  • Sentence in lines 139-140 should be omitted in Results and mentioned in methodology.

Sentences in lines 139-140 are the part of Figure 1 Title and Legend and describing in details the gating strategy. Thus, description of this figure and reference is presented in the materials and methods section.  

  • Paragraph in lines 261-268 and in lines 305-311 should also be omitted as it was explained in Material and Methods.

Paragraphs in lines 261-268 and in lines 305-311 were deleted from the Results section.

Discussion

  • Sentence in lines 483-484 is too categorical. It should only identified that other clinical situations have a Th2 response as those mentioned in the manuscript.

We rewrote this paragraph by adding information about known illnesses associated with Th2 cells.  

Correction.

Interestingly, in our study the relative number of Th2-like cells was increased in patients with severe infection. The increase in Th2-like cells was presented predominantly by CM Th cells. These findings were unexpected because the Th2 response basically reflects allergy or helminth invasion. In particular, helminthes are known to have potent immunomodulatory effects in the body through activation of anti-inflammatory Th2 immune responses which are characterized by the production of IL-4, IL-5, IL-9, IL-10 and IL-13 [45]. Besides the increased levels of Th2 cell and their cytokines were demonstrated to be crucial for the resolution of many infection diseases[46]. Patients who died from SARS showed a significant increase in the Th2 cytokines - IL-4 and IL-5[47]. Likewise, viral respiratory infections are the most common triggers of severe asthma exacerbation in children and adults [51]. Higher percentages of circulating Th2 cells were found in the patients with M.tuberculosis infection compared to the controls [48][49]. Within different autoimmune disorders the majority of patients with asthma demonstrated a predominantly Th2 immune response [50].

Sentence in lines 486-493 do not justify the severity observed in COVID-19 in relation to a polarized Th2 response so, it should be changed or omitted.

We added information about influence of Th2 polarization on disease severity.

Addition

On the other hand, higher proportions of senescent CXCR3–CCR6– Th2 were associated with death in patients with COVID-19 [52]. Furthermore, the increased levels of Th2 cell as well as their hyperactivation were closely linked with gastrointestinal symptoms, including dyspnea, hyperperistalsis and gastric fluid acidification in patients with COVID-19 [53].

  • Sentence in lines 507-508: it should be better to express T cell populations as CCR6+ and CD4+CCR6+CD161+ T cells.

We understand the concern of the Reviewer #2 regarding Th17 phenotype and thank for bringing up this important issue, we made changes to this part of Discussion

Correction

«displayed a lower levels of Th17 cells with CCR6+CD4+CD3+ and CCR6+CD161+CD4+CD3+ phenotypes» instead of «displayed a lower levels of CCR6-expressing T cells and of CCR6 and CD161 co-expressing CD4+ T cells»

 Omit sentence in lines 524-525 as other studies have shown differences in Tfh cells.

The sentence was deleted.

  • Omit sentence 543-545 because it is repetitive.

The sentence was deleted.

  • It should be recommended to mention future directions to be addressed at the end of the manuscript.

We are planning to study the cytokine profile of patients with COVID-19 and it’s dynamics  during in-patient department therapy. Also the influence of the therapy on the T- and B-cells subsets is also in a focus of our interest.

Nevertheless, as well as all these topics are in our plans and not finished yet, we think that we are not able to mention them in this manuscript.